# The Gal4-Type Transcription Factor Pro1 Integrates Inputs from Two Different MAPK Cascades to Regulate Development in the Fungal Pathogen *Fusarium oxysporum*

**DOI:** 10.3390/jof8121242

**Published:** 2022-11-24

**Authors:** Rafael Palos-Fernández, David Turrà, Antonio Di Pietro

**Affiliations:** 1Departamento de Genética, Universidad de Córdoba, 14014 Córdoba, Spain; 2Center for Studies on Bioinspired Agro-Enviromental Technology, Department of Agriculture, Università di Napoli Federico II, 80055 Portici, Italy

**Keywords:** *Fusarium oxysporum*, MAPK, Mpk1, Fmk1, Pro1, transcription factor, hyphal fusion, virulence

## Abstract

Mitogen-activated protein kinase (MAPK) signaling pathways control fundamental aspects of growth and development in fungi. In the soil-inhabiting ascomycete *Fusarium oxysporum*, which causes vascular wilt disease in more than a hundred crops, the MAPKs Fmk1 and Mpk1 regulate an array of developmental and virulence-related processes. The downstream components mediating these disparate functions are largely unknown. Here we find that the GATA-type transcription factor Pro1 integrates signals from both MAPK pathways to control a subset of functions, including quorum sensing, hyphal fusion and chemotropism. By contrast, Pro1 is dispensable for other downstream processes such as invasive hyphal growth and virulence, or response to cell wall stress. We further show that regulation of Pro1 activity by these upstream pathways occurs at least in part at the level of transcription. Besides the MAPK pathways, upstream regulators of Pro1 transcription also include the Velvet regulatory complex, the signaling protein Soft (Fso1) and the transcription factor Ste12 which was previously shown to act downstream of Fmk1. Collectively, our results reveal a role of Pro1 in integrating the outputs from different signaling pathways of *F. oxysporum* thereby mediating key developmental decisions in this important fungal pathogen.

## 1. Introduction

Signal transduction pathways control fundamental aspects of growth, development and reproduction in fungi. Among the main signaling pathways are mitogen-activated protein kinase (MAPK) cascades [1], which are present in all eukaryotic organisms and function in series to transmit a wide variety of cellular signals [2,3]. The model fungus *Saccharomyces cerevisiae* has five different MAPKs, Fus3, Kss1, Mpk1, Hog1 and Smk1 [4] while most ascomycetes possess only three MAPKs, which are orthologous to *S. cerevisiae* Kss1/Fus3, Mpk1 and Hog1 [3]. In the soil-inhabiting fungus *Fusarium oxysporum*, which causes vascular wilt on a wide variety of economically important crop plants [5], three orthologous MAPKs, Fmk1, Mpk1 and Hog1 have been reported [3]. Among these, Fmk1 and its downstream transcription factor Ste12 are essential for invasive hyphal growth and plant infection [6,7], as well as for key developmental processes such as hyphal fusion and adhesion, and for chemotropic growth towards nutrient sources [6,8,9,10,11]. Some Fmk1-controlled functions, such as pathogenicity on plants, hyphal fusion or adhesion, also depend on the cell wall integrity (CWI) Mpk1 MAPK cascade [12,13]. In addition, Mpk1 controls filamentation on a solid surface, response to cell wall stress and chemotropism towards plant root exudates and peptide pheromones [10,12,13,14]. The transcriptional regulators mediating these disparate functions downstream of the MAPK cascades are largely unknown [3].

Pro1 was originally identified in the filamentous fungus *Sordaria macrospora* as being required for the formation of sexual fruiting bodies [15]. It belongs to the Gal4 family of Zn(II)_2_Cys_6_ zinc cluster transcription factors, whose main characteristic is the presence of a conserved domain consisting of six cysteine residues which coordinate two zinc ions [16]. Pro1 orthologs were subsequently found to function in sexual development in several ascomycetes such as *Neurospora crassa*, *Cryphonectria parasitica*, *Aspergillus nidulans* and *Podospora anserina* [17,18,19,20,21,22]. Furthermore, Pro1 was shown to contribute to developmental processes that are jointly regulated by the two MAPKs Fmk1 and Mpk1 such as hyphal fusion in *N. crassa*, *Epichloë festucae* and *Aspergillus flavus* [21,23,24,25], chemotropism and cell wall remodeling in *N. crassa* [18,21] and virulence on plants in *C. parasitica*, *Alternaria brassicicola* and *A. flavus* [20,25,26]. Interestingly, the *A. flavus* and *N. crassa* Pro1 orthologs were shown to transcriptionally regulate different components of the MAK-1 (Mpk1) and MAK-2 (Fmk1) MAPK pathways [21]. Moreover, Western blot analysis suggested that the *N. crassa* Pro1 ortholog ADV-1 binds to components of the MAK-1 and MAK-2 pathways [24]. Furthermore, transcript levels of *adv-1* were markedly reduced in germlings of *mak-1*Δ and *mak-2*Δ when compared to the wild type strain, suggesting that these two MAPK pathways control expression of this transcription factor [27].

Here we investigated the role of *F. oxysporum* Pro1 in mediating a variety of responses regulated by the Fmk1 and Mpk1 MAPKs. Our findings suggest that Pro1 acts downstream of these two broadly conserved signaling cascades to control important developmental functions in this economically important fungal pathogen.

## 2. Materials and Methods

### 2.1. Fungal Isolates and Growth Conditions

The tomato pathogenic isolate *Fusarium oxysporum* f. sp. *lycopersici* 4287 (race 2; FGSC 9935) and its derivatives were used in all experiments. Fungal strains were stored at −80 °C as microconidial suspensions in 30% glycerol (*v/v*). For microconidia production, DNA extraction and fungal development, strains were grown for 3–4 days in liquid potato dextrose broth (PDB) at 28 °C and 170 rpm. For RNA extraction, freshly obtained microconidia were germinated in Puhalla’s minimal medium (MM) [28] adjusted to pH 7.4 and supplemented with 25 mM sodium glutamate and 20 mM 4-(2-hydroxyethyl)-1-piperazineethanesulfonic acid, and germlings were harvested after 10 h by vacuum filtration. Hygromycin (20 μg/mL) or neomycin G418 (10 μg/mL) were added as needed.

### 2.2. Nucleic Acid Manipulation and Quantitative Real-Time Reverse Transcription-Polymerase Chain Reaction (RT-qPCR) Analysis

Genomic DNA and total RNA were extracted from *F. oxysporum* mycelium as previously reported [9,29]. Plasmid DNA extraction was carried out as described [30]. DNA was quantified in a Nanodrop^®^ ND1000 spectrophotometer at 260 nm and 280 nm wavelengths. The quality of the DNA was monitored by electrophoresis in 0.7% agarose gels (*w/v*). PCR reactions were performed with the high-fidelity template PCR system (Roche Diagnostics) in a MJ Mini personal thermal cycler (Bio-Rad).

To measure transcript levels of the *pro1* gene, total RNA was isolated from snap frozen tissue of three biological replicates and used for reverse transcription quantitative PCR (RT-qPCR) analysis as described [9,31]. Briefly, RNA was extracted using the Tripure Reagent and treated with DNAase (both from Roche Diagnostics SL, Barcelona, Spain). Reverse transcription and qPCR were carried out with the using the FastStart Essential DNA Green Master (Roche) in a CFX Connect Real-Time System (Bio-Rad) according to the manufacturer’s instruction. Primers used for RT-qPCR analysis are listed in Appendix A. Data were analyzed using the double delta Ct method [32] by calculating the relative transcript level in relation to the *act1* gene (*FOXG_01569*).

### 2.3. Generation of Gene Deletion Mutants and Complemented Strains

Targeted deletion of the *pro1* gene (FOXG_01740) was performed by gene replacement with the hygromycin B (Hyg^r^) resistant cassette using the split marker method [33]. Briefly, two PCR fragments encompassing 1.5 kb of the 5′- and 3′-flanking regions were amplified by PCR with the primer pairs Pro1PF + Pro1 PR and Pro1TF + Pro1TR (Appendix A). The amplified fragments were then fused to the hygromycin resistance cassette, previously amplified with primers Gpda15B + Trpter8B, using the fusion primer combinations Pro1PFN/HygG and Pro1TRN/HygY, respectively. The two resulting DNA constructs were used to co-transform freshly prepared *F. oxysporum* protoplasts [6]. Obtained transformants were purified by two rounds of monoconidial isolation as described [34].

Hygromycin-resistant transformants were analyzed for targeted replacement of the *pro1* gene using PCR with the primer pairs Pro1PF + HygG and Pro1TR + HygY, as well as by Southern blot analysis with a gene-specific probe. For initial screening of transformants, colony PCR was performed using the Phire Plant Direct PCR Master Mix (Thermo Scientific™, Shanghai, China) kit according to the manufacturer’s instructions, with minor modifications [35]. Briefly, colonies were collected from transformation plates using sterile pipette tips, suspended in 20 μL kit dilution buffer, crushed with the pipette tip by pressing it against the tube wall, and incubated at room temperature for 2 h. Then the tubes were centrifuged for 2 min at 13,400 rpm and 0.5 μL of the supernatant was used as a template for a 20 μL PCR reaction. PCR amplifications with the primers CloverPro1 and Pro1compRnest were performed following the manufacturer’s instructions (Appendix A), using the following protocol: initial step of denaturation (5 min, 98 °C); 35 cycles (5 s, 98 °C, 5 s, 67 °C, 20 s/kb, 72 °C); and a final extension step (1 min, 72 °C).

For complementation of the *pro1* knockout mutant, a DNA fragment with the wild type *pro1* allele encompassing the coding region plus 1577 bp 5′- and 1066 bp 3′-flanking sequence was amplified from genomic DNA. A constitutively expressed and fluorescently tagged *pro1* allele driven by the *Aspergillus nidulans gpdA* promoter [36] (*Pgpdpro1-clover*), encompassing the *pro1* coding region N-terminally fused to the GFPclover gene containing a Linker and a 3xFLAG epitope followed by 1066 bp of 3′-flanking region (Appendix A), was obtained by fusion PCR with overlapping ends of the *gpdA* promoter; the GFPclover-Linker3-3xFLAG cassette amplified from plasmid *pUC57-1XFomClover3-3XFLAG* with primer pair GpdA15B + Sv40revnest; and the *pro1* coding region with a 1066 bp downstream region amplified from gDNA with primers CloverPro1 and Pro1compR. The obtained PCR fragments were fused in a single PCR reaction with the primer pair GpdA15 nest and Pro1compRnest (Appendix A).

The resulting linear DNA fragments were used to co-transform protoplasts of the *pro1*∆#23 mutant with the neomycin resistance cassette amplified from the *NeoR* plasmid with the primer pair GpdA15B + TrpC8B [37]. Co-transformants were purified by two rounds of monoconidial isolation in presence of geneticin G418 (InvivoGen) and analyzed for genomic integration of the transforming DNA fragments using PCR with primer pairs CloverPro1 and Pro1compRnest for *pro1* and Gpda5 and Pro1Rev (Appendix A) for Pgpdpro1-clover (Appendix A).

### 2.4. Cellophane Penetration Assay

Cellophane penetration assays were performed as described [8]. Briefly, an autoclaved cellophane sheet was placed on top of a PDA plate and 5 μL of a suspension of 2 × 10^7^ microconidia mL^−1^ was spot-inoculated at the center of the plate. After 3 days incubation at 28 °C, the cellophane sheet with the fungal colony was removed carefully and the plate was incubated for an additional day at 28 °C before the presence or absence of fungal mycelium on the underlying medium was recorded. All experiments included three replicates and were performed three times with similar results.

### 2.5. Quantification of Vegetative Hyphal Fusion and Hyphal Aggregation

To quantify germling fusion events, 40 μL of a suspension of 7 × 10^7^ mL^−1^ freshly collected microconidia was spread on top of a plate containing 5 mL of water agar (2%, *w/v*) supplemented with 25 mM NaNO_3_ with the help of 10 sterile glass beads (0.5 cm diameter) and the plate was incubated for 15 h at 28 °C. Hyphal fusion events were counted in an Olympus BH-2 microscope (Olympus Iberia, Barcelona, Spain) using differential interference contrast imaging (400× magnification). The number of germ tubes undergoing hyphal fusion was expressed as percentage of the total number of counted hyphae. For each strain, a total 300 germlings were examined. Experiments were performed at least three times. Statistical analysis was conducted using one-way ANOVA followed by Tukey’s multiple comparison performed using GraphPad Prism version 8.0.1 for Windows, GraphPad Software, San Diego, California USA, www.graphpad.com (accessed on 6 December 2021).

For macroscopic analysis of hyphal aggregate formation, 4.76 × 10^6^ mL^−1^ freshly collected microconidia were inoculated in MM supplemented with 25 mM NaNO_3_. After 36 h at 28 °C and 170 rpm, hyphal aggregates were imaged using a SteReo Lumar V12 stereomicroscope equipped with an AxioCam MR5 camera (Carl Zeiss, Barcelona, Spain).

### 2.6. Quantification of Microconidial Germination and Hyphal Chemotropism

To quantify cell-density-dependent repression of conidial germination, 10^9^ mL^−1^ freshly obtained microconidia were washed with sterile double-distilled water, transferred to MM adjusted to pH 5.0 and supplemented with 0.1% (*w/v*) sucrose to obtain a final concentration of either 3.2 × 10^5^ mL^−1^ (low density) or 8.6 × 10^7^ conidia mL^−1^ (high density), and incubated for 13 or 15 h, respectively, at 28 °C and 170 rpm. The percentage of germinated conidia was determined using an Olympus BH2 microscope with differential interference contrast imaging at 400× magnification. At least 300 conidia were examined for each isolate and experimental condition, and all experiments were performed at least 3 times. Statistical analysis was conducted using *t* test with Welch’s correction performed using GraphPad Prism version 8.0.1 for Windows, GraphPad Software, San Diego, California USA, www.graphpad.com (accessed on 6 December 2021).

Quantification of hyphal chemotropism was performed as previously described [10]. For each condition, 4–5 independent batches of cells (*n* = 100–150 cells per batch) were scored. Experiments were performed at least twice. Statistical analysis was conducted using Yates’ corrected chi-squared test (two-sided) performed using GraphPad Prism version 8.0.1 for Windows, GraphPad Software, San Diego, California USA, www.graphpad.com (accessed on 6 December 2021).

### 2.7. Colony Growth Assays

For phenotypic analysis of colony growth, 5 μL of serial dilutions (2 × 10^8^, 2 × 10^7^, 2 × 10^6^ and 2 × 10^5^ mL^−1^) of freshly obtained microconidia were spotted onto 50 mM 2-(N-morpholino)ethanesulfonic acid (MES)-buffered YPD agar plates (YPDA-MES) at pH 6.5. For cell wall and oxidative stress assays, Congo Red (CR) prepared in water (final concentration 50 μg/mL), Calcofluor white (CFW) prepared in 0.5% *w/v* KOH and 83% *v/v* glycerol (final concentration 40 μg/mL), menadione in ethanol (final concentration 10 μg/mL) and H_2_O_2_ (final concentration 0.8 mM) (all from Sigma–Aldrich, St. Louis, MO, USA) were added to the YPDA-MES medium [12]. Plates were incubated for 2 days at 28 °C and imaged, except for heat stress assays in which they were incubated for 4 days at 34 °C. The colony area was measured using ImageJ software (U. S. National Institutes of Health, Bethesda, MD, USA). Experiments were performed at least three times with similar results. The data presented are from one representative experiment. Statistical analysis was conducted using one-way ANOVA followed by Tukey’s multiple comparison performed using GraphPad Prism version 8.0.1 for Windows, GraphPad Software, San Diego, CA, USA, www.graphpad.com (accessed on 6 December 2021).

### 2.8. Tomato Plant Infection Assay

Tomato root infection assays were performed as described [33]. Briefly, two-week-old tomato seedlings (cultivar Moneymaker) were inoculated with *F. oxysporum* strains by immersing the roots in a suspension of 2.5 × 10^8^ mL^−1^ freshly obtained microconidia in water, planted in minipots with vermiculite and maintained in a growth chamber (14/10 h light/dark cycle) at 28 °C. Plant survival was recorded daily. Mortality was calculated by the Kaplan–Meier method and compared among groups using the log-rank test. All infection assays included 10 plants per treatment and were performed at least twice with similar results. Statistical analysis was conducted using the Log-rank (Mantel–Cox) test. Data were plotted using GraphPad Prism version 8.0.1 for Windows, GraphPad Software, San Diego, California USA, www.graphpad.com (accessed on 6 December 2021).

### 2.9. Sequence Retrieval and Phylogenetic Analysis

The predicted *F. oxysporum* Pro1 ortholog as well as other fungal orthologs were identified by performing a BLASTP search in the genome database of the National Center for Biotechnology Information with the *Sordaria macrospora* k-hell Pro1 amino acid sequence (NCBI database: XP_003351793.1). Amino acid sequences of the different Pro1 proteins were aligned using Clustal W [38] and manually inspected. Only fully aligned regions of the multiple sequence alignment were used. The phylogenetic tree was made using the Clustal W Modelgenerator algorithm.

## 3. Results

### 3.1. Pro1 in F. oxysporum Is under Complex Transcriptional Control by the Fmk1 and Mpk1 MAPK Cascades and the Regulators Fso1 and Velvet

A BLASTp analysis of the *F. oxysporum* genome database identified a single *pro1* orthologue, *FOXG_01740*, which encodes a predicted protein of 686 amino acids sharing a high degree of identity with Pro1 proteins from ascomycetes (Appendix A; Appendix A). To study the transcriptional control of the *pro1* gene, we measured *pro1* transcript levels by RT-qPCR analysis during early stages of germling fusion in the *F. oxysporum* wild type (wt) strain as well as in isogenic mutants lacking the MAPKs Fmk1 or Mpk1, the transcription factor Ste12 [7], the regulator of hyphal fusion Fso1 [8], the Velvet regulatory complex component VeA or its downstream regulator LaeA [39]. Compared to the wt, expression of *pro1* was moderately downregulated in the *fmk1*∆ and *ste12*∆ mutants and strongly reduced in the *mpk1*∆, *fso1*∆, *laeA*∆ and the *veA*∆ mutant (Figure 1). We conclude that *pro1* transcription during early developmental stages of *F. oxysporum* is controlled by different signaling pathways and regulators.

### 3.2. Generation of Pro1 Deletion Mutants

To examine the role of Pro1 in *F. oxysporum,* we generated *pro1*∆ mutants by replacing the entire FOXG_01740 coding region with the hygromycin resistance cassette (Appendix A). PCR analysis identified twelve hygromycin-resistant transformants producing amplification patterns indicative of homologous replacement of the *pro1* gene. Two of these transformants were further tested by Southern blot analysis, confirming the replacement of a 5196 bp *Hind*III fragment corresponding to the wt *pro1* allele, with a fragment of 9098 bp consistent with homologous insertion of the deletion construct at the *pro1* locus (Appendix A). Next, we complemented the mutant strain by re-introducing either the wt *pro1* allele *(pro1∆ + pro1)* or a constitutively expressed *pro1* allele fused to the fluorescent GFP-derivative clover and driven by the strong *Aspergillus nidulans gpdA* promoter (*pro1*∆ + *PgpdA-pro1-clover*) by co-transforming *pro1*∆ protoplasts with the neomycin resistance cassette together with a 4.7-kb DNA fragment encompassing the complete *pro1* gene, or with a construct of the *pro1* coding region fused to the clover gene and driven by the strong constitutive *A. nidulans gpdA* promoter, respectively (Appendix A). PCR analysis with gene-specific primers identified six independent transformants showing a PCR amplification product identical to that obtained from the wild type strain, which was absent in the *pro1*∆ mutants, suggesting that these *pro1*∆ + *pro1* transformants had integrated the wild type *pro1* allele (Appendix A). Furthermore, PCR analysis with specific primers for the *PgpdA-pro1-clover* construct identified 4 independent transformants carrying the constitutively expressed *pro1-clover* allele (Appendix A). RT-qPCR analysis revealed that expression of *pro1* was abolished in the *pro1*∆ knockout mutant, but only partially restored in the *pro1*∆ + *pro1* complemented strain (Figure 1). By contrast, transcript levels in the *pro1∆ + PgpdA-pro1-GFP* strain were fully restored to wt level.

### 3.3. Pro1 Is Required for Vegetative Hyphal Fusion and Hyphal Aggregation

We next examined vegetative hyphal fusion and hyphal adhesion, two processes that collectively lead to the formation of macroscopically visible hyphal networks during growth of *F. oxysporum* in liquid medium [6,8,9]. Hyphal fusion is a highly regulated developmental process required for the formation of networks and multicellular structures [40]. Here we found that approximately half of the conidial germ tubes of the *F. oxysporum* wt strain engaged in hyphal fusion whereas in the *pro1*∆ mutant hyphal fusion was largely abolished, similar to the previously described fusion-defective *fmk1*Δ and *mpk1*Δ mutants (Figure 2A) [8,12]. We also noted that hyphal fusion was fully rescued in the *pro1*∆ + *PgpdA-pro1-clover* strain but only partially so in the *pro1*∆ + *pro1* strain.

While the wild type strain formed dense hyphal aggregates, the *pro1*∆ mutants failed to produce such networks similar to the *fmk1*∆ and *mpk1*∆ mutants (Figure 2B) [9,12]. Although hyphal fusion was reduced in *pro1*∆ + *pro1* complemented strain, aggregate formation was restored to the same extent as in the *pro1*∆ + *PgpdA-pro1-clover* strain.

### 3.4. Pro1 Contributes to Quorum Sensing during Germination of Microconidia

In *F. oxysporum*, as well as in other fungi [41,42], germination of conidia is inhibited at high inoculum concentrations, a process that is mediated by quorum sensing via the Mpk1 MAPK pathway [13]. Here we tested the role of Pro1 in this process. Germination rates of the *pro1*∆ mutants at an optimal spore concentration of 3.2 × 10^5^ mL^−1^ were similar to those of the wt (Figure 3A). By contrast, at an inhibitory concentration of 8.6 × 10^7^ mL^−1^, the germination rates of the *pro1*∆ mutants were significantly higher than those of the wt (Figure 3B), similar to those previously reported for the *mpk1*∆ mutant, which is defective in quorum sensing [13].

### 3.5. Pro1 Is Required for Chemotropic Growth towards Nutrients, Plant Chemoattractants and Peptide Pheromone

Chemotropism is the ability of directed growth towards a chemical gradient. In *F. oxysporum*, chemotropism towards nutrients such as glucose or glutamate depends on the Fmk1 MAPK cascade directed growth towards peptide sex pheromones and chemoattractants secreted by plant roots requires the Mpk1 MAPK cascade [10,11,13]. Here we found that, in contrast to the wt and the *pro1*Δ + *pro1* complemented strain, the *pro1*Δ mutants failed to respond chemotropically to the nutrient glucose, peptide α-pheromone and tomato root exudate (Figure 4). Somewhat unexpectedly, only chemotropism towards glucose, but not towards pheromone and root exudate, was restored in the strain carrying the constitutively expressed *pro1*Δ + *PgpdA-pro1-clover* strain.

### 3.6. Pro1 Does Not Contribute to Cell Wall, Oxidative and Heat Stress Responses

The CWI Mpk1 MAPK cascade mediates adaptation of *F. oxysporum* to different types of stresses [10,12]. To test the role of Pro1 in this response, growth assays were performed in the presence of the cell wall targeting compounds Congo Red and Calcofluor white, the oxidative stress generating compounds menadione and H_2_O_2_, as well as high temperature (34 °C). Comparison of the colony diameters did not reveal any significant differences between the *pro1*Δ mutants and the wt (Appendix A). By contrast, as reported previously, growth of the *mpk1*Δ mutant was strongly affected on Congo Red and Calcofluor white.

### 3.7. Pro1 Is Dispensable for Invasive Hyphal Growth and Plant Infection

In other ascomycetes, Pro1 was shown to contribute to developmental processes that are jointly controlled by the two MAPKs Fmk1 and Mpk1 [18,21,24] as well as to virulence on plant hosts [20,23,26]. In *F. oxysporum*, Fmk1 controls invasive hyphal growth, a major virulence-related function that can be defined by the capacity to penetrate across a cellophane membrane [7,8]. Here we found that the *pro1*Δ mutants were unaffected in cellophane penetration, indicating that Pro1 is dispensable for Fmk1-mediated invasive growth (Figure 5A). In line with this, the *pro1*Δ mutants caused similar levels of mortality on tomato plants as the wt (Figure 5B; Appendix A). By contrast, the *fmk1*∆ mutant was non-pathogenic as previously described [6].

## 4. Discussion

The soil-inhabiting vascular wilt pathogen *F. oxysporum* uses the Fmk1 and Mpk1 MAPK cascades to regulate a variety of developmental and stress-related processes, some of which are relevant for infection [3]. Thus, Fmk1 controls invasive hyphal growth, which is essential for pathogenicity on plant hosts, as well as hyphal fusion and nutrient chemosensing [6,9,10]. Meanwhile, the CWI MAPK Mpk1 is required for hyphal fusion, quorum sensing and chemotropic growth towards peptide sex pheromones and root exudates [10,12,13]. Moreover, this Mpk1 is essential for fungal adaptation to cell wall stress [3,12]. How these two distinct MAPK cascades regulate common developmental processes through downstream transcription factors is poorly understood. For example, the zinc finger transcription factor Ste12 was previously shown to mediate invasive growth and nutrient chemosensing, but is dispensable for hyphal fusion and aggregation, suggesting that these different Fmk1-dependent processes are controlled through distinct downstream regulators [7,10]. On the other hand, the transcription factors operating downstream of Mpk1 in *F. oxysporum* remain to be identified.

Here we functionally characterized the *F. oxysporum* ortholog of Pro1, a Zn(II)_2_Cys_6_ transcription factor known to regulate developmental processes controlled by Fmk1 and Mpk1 orthologs in different ascomycetes [43]. We show that Pro1 is essential for vegetative hyphal fusion and hyphal aggregation of *F. oxysporum*, as previously reported in *N. crassa*, *E. festucae* and *A. flavus* [21,23,24,25]. Hyphal fusion is a complex developmental process that requires independent inputs from the Fmk1 and the Mpk1 cascade. In addition, *F. oxysporum* Pro1 was also required for hyphal chemotropism, mirroring the results from *N. crassa* where the Pro1 ortholog regulates chemoattraction prior to hyphal fusion [18,21]. Importantly, we found that Pro1 is required both for nutrient chemosensing (which is regulated by the Fmk1 cascade) and for chemotropism towards pheromones and root exudates (which is under control of the Mpk1 pathway) [10,11,13]. Taken together, our result suggest that Pro1 controls developmental processes downstream of both MAPK cascades. Finally, Pro1 is also required for quorum sensing during conidial germination, a process previously shown to be mediated by the Mpk1 MAPK pathway [13].

By contrast, we found that Pro1 is dispensable for invasive hyphal growth through cellophane membranes, a key infection-related process controlled by Fmk1. In line with this, Pro1 was also dispensable for virulence of *F. oxysporum* on tomato plants. Interestingly, in the saprophytic fungus *P. anserina*, Pro1 was also dispensable for cellophane penetration [22]. This finding complements our previous study showing that Fmk1-dependent invasive growth is predominantly regulated by the homeobox transcription factor Ste12 [7]. Furthermore, we found that Pro1 is not required for the response of *F. oxysporum* to cell wall stress, which depends crucially on Mpk1 [10,12], thus providing strong evidence for the presence of an additional transcription factor mediating cell wall integrity downstream of Mpk1.

How is Pro1 regulated by different upstream components? RT-qPCR analysis showed drastically and moderately reduced transcript levels of *pro1* in *mpk1*∆ and *fmk1*∆ mutants, respectively, compared to those in the wt. This is in line with a previous report on *N. crassa* [27], but contrasts with results from *P. anserina* where *pro1* transcript levels were unaltered in mutants lacking either the Fmk1 or the Mpk1 orthologs [22]. The discrepancy could be due to differences in the developmental stage used for transcriptional analysis among these studies. Furthermore, we found that mutants lacking VeA or LaeA exhibited a strong decrease in *pro1* transcript levels. LaeA was previously reported to contribute to transcriptional regulation of *pro1* in *A. flavus* [25]. Interestingly, both VeA and LaeA function in the Velvet regulatory complex which controls fungal development and secondary metabolism [44,45,46], thus adding an additional layer of transcriptional regulation of *pro1*. Finally, a *F. oxysporum* mutant in the regulatory protein Fso1 also exhibited strongly reduced *pro1* transcript levels, suggesting a possible role of Pro1 downstream of Fso1. Taken together, these results suggest a combinatorial transcriptional regulation of the *F. oxysporum pro1* gene by at least three upstream pathways (Figure 6). Further evidence for the relevance of transcriptional control for Pro1 function comes from the finding that the *pro1*∆ + *pro1* complemented strain, in which the *pro1* transcript levels were only partially restored compared to the wt (Figure 1), did not fully complement some of the phenotypes such as hyphal fusion (see Figure 2A). On the other hand, constitutive expression of *pro1* by the *gpdA* promoter failed to restore chemotropism towards peptide pheromone or plant root exudates. Taken together, these results suggest that a tight regulation of *pro1* at the transcriptional level is essential for its correct function in *F. oxysporum*.

In summary, we demonstrate here that the Zn(II)_2_Cys_6_ transcription factor Pro1 functions as a key regulator of developmental processes downstream of a variety of MAPK and other signaling cascades. It remains to be determined whether Pro1 activity by some of the upstream components is also controlled via additional mechanisms, such as phosphorylation and/or protein-protein interactions. Further studies are required to fully elucidate the complex interplay of Pro1, Ste12 and other transcriptional regulators in the control of development and pathogenicity of *F. oxysporum*.

## Figures and Tables

**Figure 1 jof-08-01242-f001:**
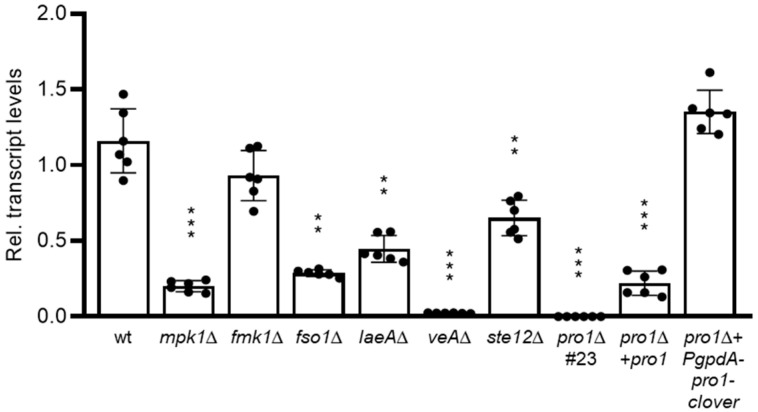
Transcriptional analysis of *pro1* in different *F. oxysporum* strains. Quantitative real-time RT-PCR analysis was performed in the indicated strains germinated for 10 h in Puhalla minimal medium supplemented with 25 mM sodium glutamate and 20 mM 4-(2-hydroxyethyl)-1-piperazineethanesulfonic acid and adjusted to pH 7.4. Transcript levels of the *pro1* gene are expressed relative to those of the wild type strain (wt). Data show means ± s.d. **, *p* < 0.001; ***, *p* < 0.0001 versus wt according to *t* test with Welch’s correction.

**Figure 2 jof-08-01242-f002:**
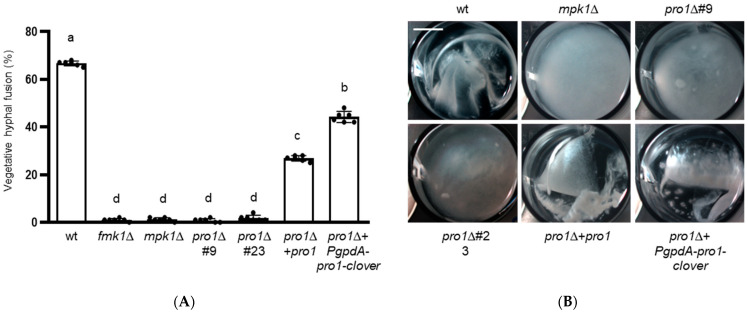
Pro1 is required for vegetative hyphal fusion and mycelial aggregation. (**A**) Microconidia of the indicated strains were germinated on water agar plates supplemented with 25 mM NaNO_3_. After 15 h the percentage of germ tubes undergoing vegetative hyphal fusion was determined. Data show means ± s.d. Columns with the same letter are not significantly different according to one-way ANOVA followed by Tukey’s multiple comparison test (*p* < 0.05). (**B**) Formation of hyphal aggregates after 36 h growth in minimal medium supplemented with 25 mM NaNO_3_. Fungal cultures were vortexed to dissociate weakly adhered hyphae, transferred to a multiwell plate and imaged under a Lumar V12 stereomicroscope equipped with an AxioCam MR5 camera.

**Figure 3 jof-08-01242-f003:**
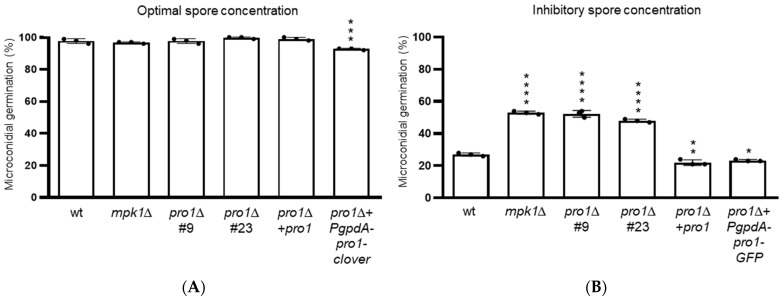
Pro1 controls quorum sensing of *F. oxysporum* microconidia. The percentage of germinated microconidia of the indicated strains at optimal (3.2 × 10^5^ mL^−1^) (**A**) or inhibitory microconidial concentrations (8.6 × 10^7^ mL^−1^) (**B**) was determined after 13 or 15 h, respectively, of incubation in minimal medium supplemented with 0.1% (*w/v*) sucrose and adjusted to pH 5.0. Data show means ± s.d. *, *p* < 0.05; **, *p* < 0.01; ***, *p* < 0.001; ****, *p* < 0.0001 versus wt according to one-way ANOVA followed by Tukey’s multiple comparison test.

**Figure 4 jof-08-01242-f004:**
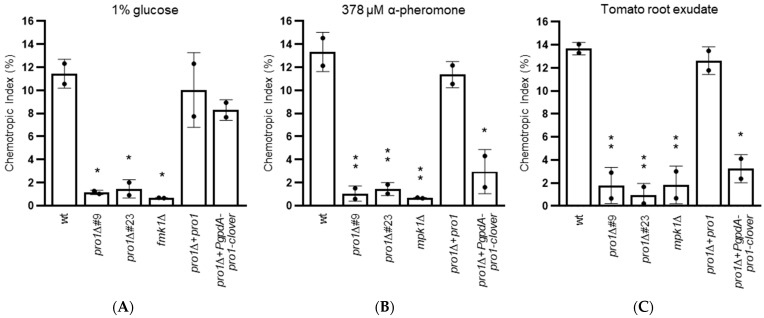
Pro1 is required for hyphal chemotropism towards nutrients (**A**), sex pheromone (**B**) and plant roots (**C**). Directed growth of germ tubes of the indicated strains towards a gradient of indicated chemoattractants was determined. Data show means ± s.d. *, *p* < 0.05; **, *p* < 0.01 versus wt according to Yates’ corrected chi-squared test (two-sided).

**Figure 5 jof-08-01242-f005:**
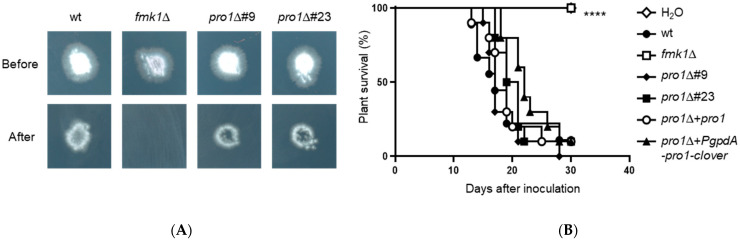
Pro1 is dispensable for invasive hyphal growth and virulence of *F. oxysporum* on tomato plants. (**A**) Cellophane penetration assay. The indicated strains were spot-inoculated on top of a cellophane membrane on a PDA plate, grown for 3 days at 28 °C and imaged (Before). The cellophane with the fungal colony was removed and plates were incubated for an additional day to determine the presence of mycelial growth on the plate, indicating penetration of the cellophane (After). (**B**) Tomato root infection assay. Kaplan–Meier plot showing survival of groups of 10 tomato plants (cv. Monica) inoculated by dipping roots into a suspension of 5 × 10^6^ microconidia/mL of the indicated fungal strains. Percentage survival was plotted for 30 days (****, *p* < 0.0001). Data shown are from one representative experiment. All experiments were performed at least three times with similar results.

**Figure 6 jof-08-01242-f006:**
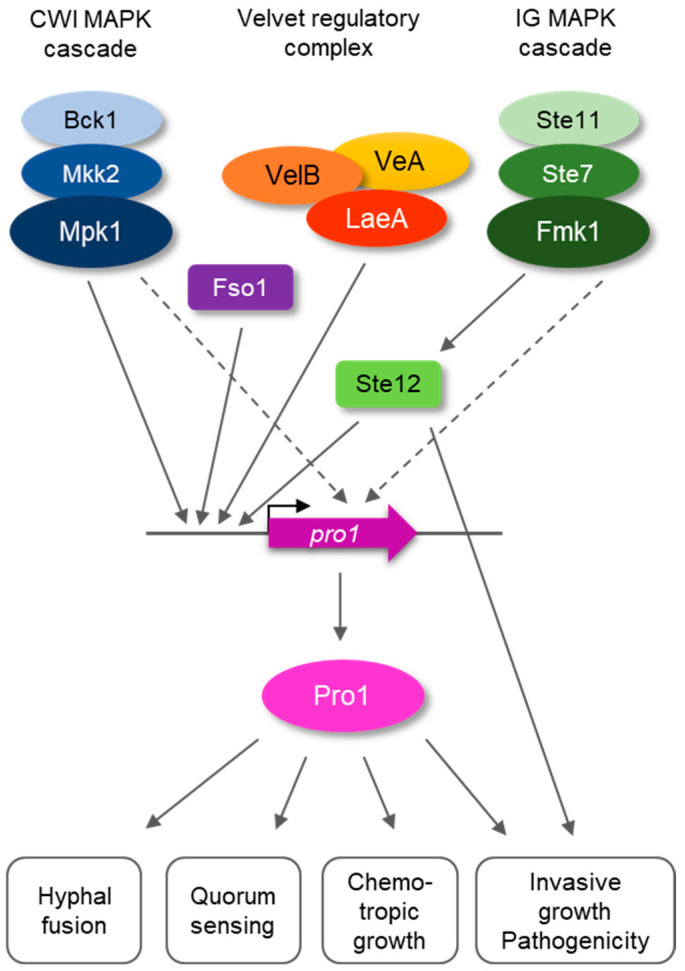
Diagram showing regulation of Pro1 by different upstream components in *F. oxysporum*.

## Data Availability

Not applicable.

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
