# Peer review of "The Gal4-Type Transcription Factor Pro1 Integrates Inputs from Two Different MAPK Cascades to Regulate Development in the Fungal Pathogen Fusarium oxysporum"

_jof, 2022, doi:10.3390/jof8121242_

Round 1

Reviewer 1 Report

This interesting manuscript presents experimental data on functional characterization of the GATA-type transcription factor Pro1 in Fusarium oxysporum, a fungal pathogen causing vascular wilt disease of many crops. The authors constructed and analyzed targeted gene deletion and complementation mutants. Their data demonstrate that Pro1 is able to integrate signals from the signaling cascades of two MAPKs (Fmk1 and Mpk1) and hence involved in orchestrating quorum sensing, hyphal fusion and chemotropism of F. oxyspoorum. The whole study was carried out in a technically sound manner, and data were analyzed and presented properly. The manuscript was well written. I recommend the manuscript to be accepted for publication in JoF after a minor revision.

 Minor:

1.         Repeated proof reading is need to delete some typos or insert a space between words or between a value and a unit of volume or weight.

2.         In Figure 1, the presented error bars look like standard deviations (SD) rather than standard errors (SE). Error bars also should be clarified to be SD or SE in the bar charts of other figures.

3.         In Figure 2A, the order of lowercase letters marked for statistical significances among tested strains needs a correction, namely a for WT, b and c for two complemented strains, and d for four deletion mutants.

4.         The tested strains' differences shown in Figure 3 can be better differentiated by Tukey's multiple comparison than by T test.

5.         Three bar charts in Figure 4 should b organized together.

Author Response

We are grateful to the reviewer for the thorough revision and the constructive comments and criiticsm. We have addressed all the points raised by the reviewer.

Reviewer 2 Report

The authors investogated the role of GATA-type transcription factor Pro1 in MAPK pathways of F. oxysporum signaling pathways. The results showed that Pro1 controls a subset of functions, including quorum sensing, hyphal fusion and chemotropism.

The manuscript provided new knowledge about role of transcription factor Pro1 in gene transcription regulation.

I have the following comments:

Line 101/102: Delete respectively

Line 115: Delete (RT)

Line 164: Insert space: 25mM

Line 172: Add a comma: ..,respectively, at 28°C and 170 rpm.

Line 193: Insert space: 0.8mM

Line 291: Insert space: 25mM

References

Line 451-460: Delete the authors' instructions

Supplementary Figure 1. Delete respectively: Asterisks, double dots, and single dots indicate highly, less highly and moderately conserved residues, respectively.

Author Response

(The authors gave the same response as above.)

Reviewer 3 Report

In this manuscript the authors investigate the role of Pro1 transcription factor in regulating various developmental processes in Fusarium oxysporum. Through different set of experiments they conclude that Pro1 is regulated by atleast 3 upstream pathways. Whereas quorum sensing, hyphal fusion and chemotropism are controlled by Pro1, it is dispensable for invasive hyphal growth and virulence.

1. The biggest critique for this paper is the erratic findings of the complements. For Figure 1 and 2 the regular complement doesn't mirror the wild type but constitutive does. For Figure 3, both the complements behave like wild type and for chemotropism experiment, it is all over the place. It is understandable how perfect complementation sometimes does not happen with all the sequence disruptions, but the findings lack adequate explanation.

2. For Fig 5A, it would be scientifically sound to include findings from the complement strains as well.

3. Findings with respect to quorum sensing at high spore concentration needs additional explanation. 

Author Response

  1. The biggest critique for this paper is the erratic findings of the complements. For Figure 1 and 2 the regular complement doesn't mirror the wild type but constitutive does. For Figure 3, both the complements behave like wild type and for chemotropism experiment, it is all over the place. It is understandable how perfect complementation sometimes does not happen with all the sequence disruptions, but the findings lack adequate explanation.

We thank the reviewer for this valuable suggestion. We have followed this suggestion and introduced explanations in the relevant sections of Results and Discussion, on the partial complementation phenotypes depending on the level of pro1 expression.

  1. For Fig 5A, it would be scientifically sound to include findings from the complement strains as well.

While we are grateful for the suggestion, we consider that in this case there is no need to include the complemented strains since no differences were detected between the wt and the pro1 null mutant.

  1. Findings with respect to quorum sensing at high spore concentration needs additional explanation. 

Thank you for the suggestion. We have added a more detailed comment in the relevant Discussion section.